# Multicentre longitudinal study of fluid and neuroimaging BIOmarkers of AXonal injury after traumatic brain injury: the BIO-AX-TBI study protocol

Neil Samuel Nyholm Graham [1,2] Karl A Zimmerman,[1,2] Guido Bertolini,[3] Sandra Magnoni,[4] Mauro Oddo,[5] Henrik Zetterberg,[6,7] Federico Moro,[3] Deborah Novelli,[3] Amanda Heslegrave,[8] Arturo Chieregato [9] Enrico Fainardi,[10] Joanne M Fleming,[3] Elena Garbero [3] Samia Abed-Maillard,[5] Primoz Gradisek,[11] Adriano Bernini,[5] David J Sharp[1,2]

NSNG and KAZ are joint first authors.

For numbered affiliations see end of article.

**Correspondence to**
Professor David J Sharp;
david.sharp@imperial.ac.uk

## ABSTRACT

**Introduction and aims** Traumatic brain injury (TBI) often results in persistent disability, due particularly to cognitive impairments. Outcomes remain difficult to predict but appear to relate to axonal injury. Several new approaches involving fluid and neuroimaging biomarkers show promise to sensitively quantify axonal injury. By assessing these longitudinally in a large cohort, we aim both to improve our understanding of the pathophysiology of TBI, and provide better tools to predict clinical outcome.

**Methods and analysis** BIOmarkers of AXonal injury after TBI is a prospective longitudinal study of fluid and neuroimaging biomarkers of axonal injury after moderate-to-severe TBI, currently being conducted across multiple European centres. We will provide a detailed characterisation of axonal injury after TBI, using fluid (such as plasma/microdialysate neurofilament light) and neuroimaging biomarkers (including diffusion tensor MRI), which will then be related to detailed clinical, cognitive and functional outcome measures. We aim to recruit at least 250 patients, including 40 with cerebral microdialysis performed, with serial assessments performed twice in the first 10 days after injury, subacutely at 10 days to 6 weeks, at 6 and 12 months after injury.

**Ethics and dissemination** The relevant ethical approvals have been granted by the following ethics committees: in London, by the Camberwell St Giles Research Ethics Committee; in Policlinico (Milan), by the Comitato Etico Milano Area 2; in Niguarda (Milan), by the Comitato Etico Milano Area 3; in Careggi (Florence), by the Comitato Etico Regionale per la Sperimentazione Clinica della Regione Toscana, Sezione area vasta centro; in Trento, by the Trento Comitato Etico per le Sperimentazioni Cliniche, Azienda Provinciale per i Servizi Sanitari della Provincia autonoma di Trento; in Lausanne, by the Commission cantonale d'éthique de la recherche sur l'être humain; in Ljubljana, by the National Medical Ethics Committee at the Ministry of Health of the Republic of Slovenia. The study findings will be disseminated to patients, healthcare professionals, academics and policy-makers including through presentation at conferences and peer-reviewed publications. Data will be shared with approved researchers to provide further insights for patient benefit.

**Trial registration number** NCT03534154.

## Strengths and limitations of this study

► The BIOmarkers of AXonal injury after traumatic brain injury (TBI) observational study will characterise relationships of novel fluid and neuroimaging biomarkers of axonal injury to clinical outcomes over 12 months after moderate-to-severe TBI.

► Our study operates across multiple European centres with a core programme of work (including blood biomarkers and baseline MRI) supplemented in selected research sites by longitudinal MRI, advanced imaging sequences and cerebral microdialysis.

► We will assess whether biomarkers such as plasma neurofilament light, MRI brain atrophy and white matter damage are suitable surrogate outcome measures for possible use in clinical trials.

► We will use common data elements to capture clinical, demographic and injury information, our data will be comparable across other large cohort studies of TBI validating our results in the international context.

► In common with any longitudinal observational study there is a risk of lost to follow-up, which we assume and account for in our sample size calculations.

## INTRODUCTION

Traumatic brain injury (TBI) often results in persistent disability due particularly to cognitive and psychiatric impairments.[1] The impact of these problems on everyday function and prediction of long-term outcomes is often assessed inadequately by standard clinical assessment and imaging techniques. Most survivors are young and have near-normal life expectancy.[2] Hence, the burden on public

health and social care is substantial.[3] Improved methods of tracking progress and predicting outcomes are needed following TBI. These include clinical, neuropsychological, fluid biomarker and neuroimaging measures that could potentially be combined to yield important information. The proposed work aims to investigate clinical outcomes after TBI and identify the best way to track progress and predict outcomes.

Diffuse axonal injury after TBI results in damage to white matter connections and disruption of brain network structure and function. These changes are known to correlate with long clinical outcomes.[4 5] Previously, it has been difficult to study the location and extent of this damage or its functional consequences. Now, using new multimodal techniques, the consequences of TBI on brain connectivity can be defined more clearly. Diffusion tensor imaging (DTI) allows structural damage to white matter connections to be identified after TBI. This technique allows us to assess whether damage to specific tracts results in a predictable cognitive deficit.[6 7]

In addition to structural brain imaging, functional MRI (fMRI) can be used to provide complementary information about the consequences of white matter damage at different spatiotemporal scales following TBI. We have used this approach to provide important insights into the biological basis of cognitive impairment after TBI. FMRI studies of TBI have consistently demonstrated abnormal task-related activation within frontal and temporal lobes.[8 9] Understanding the effects of axonal injury on functional outcomes is essential to identify and predict cognitive impairment following TBI. We will use advanced MRI methods to investigate how best to identify the structural and functional effects of diffuse axonal injury and their relationship to clinical outcomes.

Fluid biomarkers provide a complementary measure of axonal injury after TBI. Damage to neurons and glial cells can lead to increases in fluid biomarkers that are clinically informative.[10–12] For example, cytoskeletal proteins are released from damaged axons and can be detected in the blood.[13 14] Recent advances have identified a number of promising biomarkers with markedly increased sensitivity making it feasible to investigate and validate a blood biomarker of axonal injury.[15] In this study, we will investigate the relationship between plasma and neuroimaging markers of axonal injury following TBI to determine whether a blood biomarker of axonal injury is valid and also study the time course of cognitive changes after TBI and their relationship to underlying brain injury.

Cerebral microdialysis can provide particularly important insights into the origin of biomarkers for axonal injury. Cytoskeletal proteins from damaged axons are released into the cerebral interstitial fluid compartment during acute axonal injury, from where they can be directly sampled.[16 17] This can be achieved with high temporal resolution, providing a method for assessing the effects of therapeutic interventions.[18] Recent findings indicate that microdialysis-based measurement of tau and neurofilament light (NFL) in the brain extracellular space may be a useful way to assess the severity of axonal injury in the acute setting. High initial microdialysis levels of tau are correlated with worse clinical outcomes.[16] In addition, microdialysis concentrations of tau are proportional to abnormalities in diffusion tensor MRI, indicating that both measures reflect the same underlying axonal injury.[17] In contrast, the relationship between local NFL concentrations and axonal injury has yet not been explored, although our preliminary data indicate that brain extracellular levels of NFL remain elevated for a longer period than tau and show heterogeneous dynamics that may reflect the progression of underlying axonal injury.

The study links closely with a multicentre European collaboration of clinical outcomes after TBI (Collaborative REsearch on ACute Traumatic Brain Injury in intensiVe Care Medicine in Europe, 'CREACTIVE', NCT02004080). Integrating with this project allows us to collect common data across a large number of patients with TBI, which allows work to be validated in relation to international experience.[19 20] CREACTIVE is part of the International Initiative for Traumatic Brain Injury Research, a large international collaborative effort to advance clinical TBI research. We will particularly integrate neuroimaging, blood biomarker and neuropsychological assessments. Quality of life and clinical measures will also be used in order to identify functional outcomes of patients during the tracking of their cognition.

## GOALS AND OBJECTIVES OF BIOMARKERS OF AXONAL INJURY AFTER TBI

### Acute injury measures and relationship to clinical outcome (work package 1)

We aim to identify the most informative plasma biomarker(s) of the severity of axonal injury using a large multicentre cohort of adult moderate-to-severe patients with TBI. We will characterise their time course focusing on NFL and tau, and relate these to MRI measures of axonal injury in the early phase postinjury. We will assess how these measures contribute to the prediction of clinical, cognitive and functional outcome at 12 months.

### Longitudinal neuroimaging and detailed cognitive assessment (work package 2)

In a subgroup of the recruited patients, we will use advanced MRI and longitudinal assessments to provide a more detailed description of the relationship between the plasma biomarkers and outcome after TBI. We will test whether advanced diffusion measures correlate with plasma biomarkers and whether early plasma biomarker levels predict neurodegeneration measured by progressive atrophy after TBI.

### Microdialysis assessment of biomarker dynamics (work package 3)

In a second subgroup of patients, we will combine microdialysis, neuroimaging and plasma sampling of axonal

proteins to provide a deeper understanding of the mechanisms of axonal injury progression and use this approach to investigate the axonal origin of the plasma biomarkers.

## Validation in large cohort and CT head neuroimaging analysis (work package 4)

The outputs of the above will be used to select the plasma biomarkers of axonal injury that best predict clinical outcome. These biomarkers will be validated by exploiting a large sample (~n=1000) of patients with TBI collected within the CREACTIVE project. In addition, the relationship between the CT head scan appearances and plasma biomarkers will be investigated using machine learning analysis to test whether CT head scans contain specific features of axonal injury and whether these features can be used to help predict outcome.

## METHODS AND ANALYSIS
### Overall design

BIO-AX-TBI is a prospective multicentre observational study of TBI clinical outcomes. Patients with acute moderate-to-severe TBI, as per the Mayo Classification of injury severity,[21] will be recruited from trauma centres taking part in the study and followed up longitudinally over a year, with assessments taking place acutely, at 10 days – 6 weeks, 6 months and 12 months postinjury.

## Participating centres and recruitment progress

BIO-AX-TBI started recruitment in 30 November 2017 and involves eight different trauma centres across Europe. These include: Lausanne University Hospital, Switzerland, St George's and St Mary's Hospitals, London, University Medical Centre, Ljubljana, Slovenia, and, in Italy, Carregi University Hospital, Santa Chiara Hospital, Trento, Niguarda Hospital and Policlinico in Milan.

Our targets are to recruit at least 250 patients in total, including 40 with invasive cerebral microdialysis. As of June 2020, 311 patients after moderate-to-severe TBI have been recruited into the study (figure 1), with recruitment ongoing in order to meet our target in the cerebral microdialysis group.

## Entry into the study

Patients suitable for the research will be identified by the clinicians and research team working with the patients in

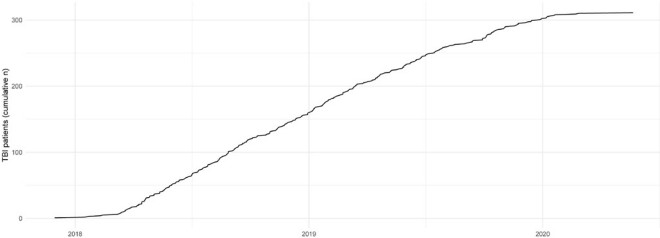

**Figure 1** Recruitment status. Cumulative number of patients recruited into the BIO-AX-TBI study since its initiation, a total of 311 participants as of June 2020. BIO-AX-TBI, BIOmarkers of AXonal injury after traumatic brain injury.

| Table 1 | Inclusion and exclusion criteria |
| --- | --- |
| **Inclusion criteria** | **Exclusion criteria** |
| Moderate-to-severe TBI (Mayo classification) | For whole study: |
| Age 18–80 | Previous significant TBI (requiring hospitalisation), |
| | Moribund patients |
| | Cardiac arrests |
| | Inability or unwillingness to participate in study |
| | Prior significant neurological or psychiatric condition |
| | Previous significant disability from any cause |
| | For MRI: typical MRI contraindications of ferromagnetic implants in the body, claustrophobia, pregnancy |

TBI, traumatic brain injury.

trauma centres across participating sites. Those satisfying the inclusion and exclusion criteria will be approached and given written and verbal information about the study and invited to participate (table 1).

Moderate-to-severe injuries will be identified as per the Mayo classification, if any of the following features are present: death due to the TBI, loss of consciousness of more than 30 min, post-traumatic amnesia (PTA) duration of more than 24 hours, lowest Glasgow Coma Scale of less than 13 in the first 24 hours, imaging abnormalities such as intracerebral haematoma, subdural, extradural, contusion, penetrating injury of dura, subarachnoid haemorrhage or brainstem injury.[21]

Individuals will have the opportunity to ask questions about the study. Questions will be asked to assess the individual's suitability for the study. Potential participants will be asked to consider the request for a period of 24 hours prior to recruitment into the study. If the potential participant is suitable for the study, informed consent will be obtained. Our goal is to recruit patients as early after trauma as is practical, and ideally within ten days of injury in order to facilitate acute blood biomarker assessment.

If a patient is unable to provide fully informed consent, we will assent with permission from next of kin or personal/nominated consultee, according to the national legislations. Patients that have been assented will be re-consented if they are subsequently able to provide full informed consent. The consent procedure will be carried out in strict compliance with related local/national legislation and, where applicable with General Data Protection Regulation (GDPR). Any waivers of consent provided for by national public health legislation, for example, where this is in the public interest, will be indicated. Subjects will be free to withdraw, or be withdrawn by their legal

**Table 2** Work packages in participating major trauma centres

| Institution | City | Country | Work package |
|---|---|---|---|
| Careggi University Hospital | Florence | Italy | 1 |
| Lausanne University Hospital | Lausanne | Switzerland | 3 |
| Niguarda Hospital | Milan | Italy | 1 |
| Policlinico of Milan | Milan | Italy | 2 |
| Santa Chiara Hospital | Trento | Italy | 1 |
| St Mary's Hospital | London | UK | 2 |
| St George's University Hospital | London | UK | 2 |
| University Medical Centre | Ljubljana | Slovenia | 1 |

representative if appropriate, at any point in the study, and they need not state any reason.

### Structure of assessments

The study is organised into separate work packages, reflecting differing longitudinal assessment programmes for those at recruiting sites (table 2), as set-out in 'goals and objectives'.

Patients participating in work package 1 will have clinical and demographic information collected acutely using the electronic case report form (eCRF). Blood samples will be collected twice in the first 10 days after injury, and at subsequent visits at 10 days – 6 weeks, and around 12 months. Standard MRI scanning will be performed at 10 days to 6 weeks only. Outcomes will be assessed at 6 months via telephone and 12 months via a face-to-face assessment. If face-to-face follow-ups are not possible then remote follow-up will be attempted for example, through telephone consultation.

Work package 2 comprises a more detailed programme of assessment. As per WP1, the eCRF will be completed acutely alongside blood tests twice in the first 10 days after injury. Advanced MRI will be performed at the 10 days to 6 weeks study visit, alongside blood biomarker assessment and clinical/cognitive outcome assessment. Blood biomarker, advanced MRI and outcome assessment will be repeated face to face at 6 and 12 months after injury.

Work package 3 involves the addition of cerebral microdialysis assessments acutely during the inpatient admission. Blood biomarker assessment is more frequent, up to twice per 24 hours period during cerebral microdialysis. Standard MRI is performed alongside blood biomarkers at 10 days to 6 weeks. Face-to-face outcome assessments are performed at 6 and 12 months postinjury.

Work package 4 is an analysis-only package involving comparing our data (WP1-3) to a broader sample of patients with TBI collected within the related CREACTIVE project, including neuroimaging data. This work package does not involve collection of new data or

biological samples. Two main analyses are planned: first, to compare clinical features, biomarkers and outcomes in patients in the BIOAXTBI cohort with the broader CREACTIVE population, where these are in-common, and establish how well findings from WP1-3 generalise in the larger group. Second, a machine learning analysis will be performed to establish whether CT brain imaging data in this large cohort can reliably predict outcomes.

Healthy volunteers will have a single time point MRI assessment and blood biomarker assessment using the same protocol as patients. Assessments of cognition, motor function, psychiatric outcome and sleep quality will be performed. Fifteen healthy control subjects per centre (total n=105) will be recruited, matched to patients with TBI for age and sex.

### Electronic case report form

Participant details will be collected using an eCRF ('Prosafe') requiring population of TBI common data elements.[19] This was developed as part of the large multicentre Gruppo Italiano per la Valutazione degli Interventi in Terapia Intensiva—Italian Group for the Evaluation of Interventions in Intensive Care Medicine intensive care unit network in Italy,[22] and provides a framework to collect high-quality records that meet strict criteria. The software facilitates data entry at research sites, pseudonymisation and centralisation across sites for analysis. Data are entered for all participants, including a limited healthy volunteer eCRF for control data collection.

### Clinical, cognitive and functional outcome assessment

Study participants will undergo a number of tests of clinical outcome and cognitive function as they recover from injury (table 3).

Standardised pen and paper neuropsychological tests and a number of questionnaires designed to assess functional outcomes after TBI will be used. Assessments will coincide with biomarker assessment timepoints. Healthy volunteer assessments will be performed to facilitate comparison (box 1).

Where appropriate, family or close friends will also be asked to complete caregiver questionnaires to provide an objective measure of perceived impairments and functional outcomes. Assessments will be performed by an appropriately trained investigator. These may include: Lille apathy ratings scale, Frontal Systems Behaviour Scale and the Mayo-Portland Adaptability Inventory 4.

### Blood biomarkers

Blood samples (≈12 mL on each occasion) will be taken on different number of occasions depending on the work package the patient is recruited to. BIO-AX-TBI samples comprise: 2x K3 EDTA (ethylenediaminetetraacetic acid) (6 mL, for plasma biomarkers) and where ethical approvals permit, 1x further K3 EDTA (6 mL, whole blood for DNA analysis). The aligned CREACTIVE study involves the following sample collection: 1 x bottle of each sodium citrate (3.5 mL), serum z-clot activator (5 mL)

**Table 3** Outcome assessments in patients after TBI

| Outcome measure | Assessment timepoint | | |
| --- | --- | --- | --- |
| Functional outcome | 10 days to 6 weeks | 6 months | 12 months |
| Glasgow Outcome Scale-Extended | WP 1,2,3 | WP 1,2,3 | WP 1,2,3 |
| Frontal Systems Behaviour Scale* | | WP 2,3 | WP 2,3 |
| Lille Apathy Ratings Scale* | | WP 2,3 | WP 2,3 |
| Mayo-Portland Adaptability Iinventory-4* | WP 2.3 | WP 2.3 | WP 2,3 |
| Cognitive function | | | |
| Stroop test (Delis-Kaplan Executive Function System) | WP 2,3 | WP 2,3 | WP 2,3 |
| Trail making tests A and B. | WP 2,3 | WP 2,3 | WP 2,3 |
| Montreal Cognitive Assessment | | | WP 1,2,3 |
| Computerised Go/No-Go, Corsi blocks and N-back | | | WP 1,2,3 |
| Repeatable Battery for the Assessment of Neuropsychological Status | WP 2,3 | WP 2,3 | WP 2,3 |
| Motor function | | | |
| Box and block test of motor function | | | WP 1,2,3 |
| Functional ambulatory category questionnaire | | | WP 1,2,3 |
| Quality of life | | | |
| Quality of Life After Brain Injury – Overall Scale | WP 1,2,3 | WP 1,2,3 | WP 1,2,3 |
| Psychiatric outcome | | | |
| Hospital anxiety and depression scale | | | WP 1,2,3 |
| Post-traumatic stress disorder checklist for DSM-V (PCL-5) | | | WP 1,2,3 |
| Sleep quality | | | |
| Insomnia Severity Index | | | WP 1,2,3 |
| Rehabilitation treatment | | | |
| Rehabilitation Pathway Questionnaire | | WP 1,2,3 | WP 1,2,3 |

*administered to patient and caregiver

DSM-V, Diagnostic and Statistical Manual of Mental Disorders, 5th Edition; PCL-5, PTSD CheckList for DSM-V; TBI, traumatic brain injury.

---

**Box 1  Cognitive and functional assessments in healthy volunteers**

**Cognitive function**
Stroop test (Delis-Kaplan Executive Function System).*
Trail making tests A and B.*
Montreal cognitive assessment.
Computerised Go/No-Go, Corsi blocks and N-back.

**Motor function.**
Box and block test of motor function.
Functional Ambulatory Category Questionnaire.

**Psychiatric outcome.**
Hospital anxiety and depression scale.

**Sleep quality.**
Insomnia Severity Index.
*Excluding centres performing WP1 only.

---

and K3 EDTA (6 mL). BIO-AX-TBI samples are taken twice in the first 10 days after TBI. In those participants undergoing microdialysis, a K3 EDTA 6 mL sample may be taken up to a maximum of 12 hourly for 14 samples. Sample processing involves centrifugation at 2000–2500 g for 10–20 min at 4°C.

Blood and other human samples (ie, microdialysis samples) will be labelled using a human readable barcode system. We will process, store and dispose of all tissue in accordance with all applicable legal and regulatory requirements. All blood samples will be processed within 2 hours of collection. Microdialysis samples will be stored at local facilities and sent to the central lab afterwards. Patient identity is kept confidential.

Samples will be kept in locked storage and may be stored for up to 15 years from the end of the study. Participants can withdraw their consent for their samples to be stored or used at any point. If they choose to do so, and notify us, their samples will not be used for any further research and will be destroyed in accordance with all applicable legal and regulatory requirements. Analysis of blood samples will allow circulating factors related to outcomes after TBI to be identified. These may include: NFL, tau, glial fibrillar acidic protein, ubiquitin C-terminal hydrolase 1 and protein S100 (S100B). Fluid biomarker analyses will be performed at University College London (UCL) using a digital ELISA technique, using a Quanterix Simoa analyser to provide ultrasensitive assessment of concentrations.

Anonymous samples may be analysed either within Imperial College London (ICL), UCL or in other institutions or laboratories, including those outside the UK and EU. In participating centres where the relevant ethical approvals are granted, we will process the blood samples to allow for assessment of known genetic factors and factors that will be eventually known in the future that may influence clinical outcome after TBI. We will store DNA samples without any personally identifiable information for future analysis. Patients will not be informed of the results of genetic analysis. These DNA samples will be stored in an anonymised fashion in a secure −80°C freezer. Genetic analyses may include Apolipoprotein E (APOE) genotype assessment and/or generation of an individualised polygenic risk score for neurodegeneration, using

a microarray targeting single nucleotide polymorphisms (SNPs) related to neurodegenerative disease.[23]

## Cerebral microdialysis

Patients will be monitored with cerebral microdialysis as part of standard patient care and according to local protocols.[24] This will start immediately after injury and continue until clinically indicated (normally 5–7 days). The 100 kD cut-off MDialysis AB microdialysis catheters will be surgically implanted through a burr hole in white matter regions that appear normal on CT (mainly in the right, non-dominant frontal lobe). In patients with frontal contusions, catheters will be placed distant to intracerebral lesions. A standard 2 mm non-contrast brain CT scan will be taken within 24 hours to verify correct catheter placement. CT scan images will be coregistered with MRI.

Colloids such as dextran or albumin solutions will be added to the standard perfusion fluid to increase both fluid and protein recovery, as indicated by the manufacturer, and as recently reported.[25] A standard flow rate of 0.3 µL/min will be used in all patients, allowing hourly recovery of brain extracellular fluid (about 20 µL). A subset of samples will be routinely analysed for lactate, pyruvate, glycerol, glutamate and glucose at bedside using a portable analyser (Iscus Flex, MDialysis AB, Stockholm, Sweden) to measure in vivo changes of cerebral metabolites. These data will be displayed real time for clinical use as indicated by published guidelines (16). Six vials (four hourly during collection) will be immediately frozen and stored after collection for the measurement of biomarker concentrations (eg, tau and NFL). These samples will be centralised to UCL laboratory.

Microdialysis and brain physiology data, including mean arterial pressure, intracranial pressure, cerebral perfusion pressure and core temperature, will be continuously recorded as part of a multimodal monitoring system and transferred to excel sheets. Additional information, including lab and arterial blood gas analysis data, hourly vital signs and ventilation parameters, if applicable, will be recorded.

## MRI assessment

Participants will undergo an MRI scanning session comprising structural and fMRI scans (see table 4). The timing of imaging assessments depends on the work package in which a given participant is enrolled. Participants will undergo a set of structural MRI scans including T1 (for high-resolution detail), susceptibility-weighted imaging (SWI) and T2 fluid-attenuated inversion recovery (FLAIR) (for detecting other abnormalities), T2 mapping (for detecting oedema) and diffusion MRI (for measuring white matter and white matter damage). fMRI will be used to measure regional brain activity over time. FMRI will be performed with the subject lying at rest in the scanner (resting-state fMRI). These different types of MRI provide complementary information about the location of brain injury and the effects of these injuries on brain function and cognition.

Repeated MRI assessment are necessary as damage after TBI can be progressive and the brain scanning provides information about the timing of this progression. We will investigate how the MRI signature of axonal injury develops over time and how brain shrinkage atrophy evolves. Our aim is to use these measures as prognostic

| Table 4 | MRI assessment details | | | | | | |
|---|---|---|---|---|---|---|---|
| | **London** | **Niguarda** | **Lausanne** | **Ljubljana** | **Florence** | **Milan** | **Trento** |
| Scanner | | | | | | | |
| Manufacturer | Siemens | Philips | Siemens | Siemens | Siemens | Philips | GE |
| Model | Verio | Achieva | Skyra Fit | Trio Tim | Aera | Achieva | Optima |
| Field strength | 3T | 1.5T | 3T | 3T | 1.5T | 3T | 1.50% |
| Software | MR B17 | Rel. 5 Neuroradio | MR B19 | MR B19 | VE11a | Rel. 5 Neuroradio | DV25.1_R03_1802.a |
| Voxelsize (mm) | | | | | | | |
| MPRAGE | 1×1×1 | 1×1×1 | 1×1×1 | 1×1×1 | 1×1×1 | 1×1×1 | 0.5×0.5×1 |
| DTI | 2×2×2 | 2×2×2 | 2×2×2 | 2×2×2 | 2×2×2 | 2×2×2 | 1×2×2 |
| Rs-fMRI | 3×3×3 | 3×3×3 | 3×3×3 | 3×3×3 | 3×3×3 | 3×3×3 | 3×3×3 |
| FLAIR | 1×1×1 | 1.2×1.2×0.7 | 0.5×0.5×1 | 1×1×1 | 0.5×0.5×1 | 1×1×1 | 1.2×0.5×0.5 |
| SWI | 0.6×0.5×1.2 | 1×1×1 | 0.3×0.3×1.6 | 0.6×0.5×1.2 | 0.8×0.8×0.8 | 0.5×0.5×1.2 | 0.5×0.5×1 |
| DTI parameters | | | | | | | |
| Receiver coil channels | 32 | 8 | 64 | 32 | 20 | 32 | 8 |
| Directions | 64 | 64 | 64 | 64 | 64 | 64 | 60 |
| b value | 1000 | 1000 | 1000 | 1000 | 1000 | 1000 | 1000 |

DTI, diffusion tensor imaging; fMRI, functional MRI; SWI, susceptibility-weighted imaging.

indicators and also as imaging biomarkers to evaluate novel therapeutic interventions.

## Sample size

The primary outcome measures are (1) change in diffusion tensor MRI measures over time (time frame: 10 days to 6 weeks, 6 months and 12 months), measured using fractional anisotropy (FA); (2) brain atrophy rates (time frame: 10 days to 6 weeks, 6 months and 12 months), using Jacobian determinant (JD) atrophy rates; (3) Change in levels of fluid biomarkers in blood (time frame: 0–5 days, 5–10 days, 10 days to 6 weeks, 6 months and 12 months) (4) Change in levels of fluid biomarkers in cerebral fluid (time frame: 48 hours to 7 days).

We estimate that a minimum of 140 patients will be necessary to test the contribution of DTI to prognostic modelling (type 1 error=0.05, power=0.95). Lost to follow-up is an important but unpredictable element and we have allowed for up to a further 25%–30% lost to follow-up by aiming to enrol 250 patients in the study. Our estimates indicated that a minimum of 12 imaging controls will be necessary for each centre.

To assess atrophy progression after TBI in work package two, our calculations indicate a sample size of 10 per group is needed to detect a group difference in brain atrophy over 6 months with effect size of f=0.32, assuming 95% and a correlation between repeated measures=0.9. Longitudinal brain atrophy rates will be calculated using the JD measure of atrophy rate generated using SPM 12 (UCL).[26]

Preliminary data suggest that moderate-to-severe TBI induces large changes in blood biomarker concentrations (eg, S100B, NFL). We estimate that the numbers required to show between group differences and relationships to outcome will be smaller than for DTI (1) or brain atrophy rates (2), on whose measures the study is primarily powered.

We have based the WP3 sample size (n=40) on previous similar microdialysis studies that have used 15–20 subjects and shown correlations between biomarker and imaging measures. With a sample size of 40 we should be adequately powered for this analysis, as well as being able to take into account the additional variability related to different centres.

## Statistical analysis plan
### Fluid biomarkers

Blood biomarker trends will be described and compared between groups, as well as within individuals longitudinally. Distributions of variables will be assessed for normality and parametric/non-parametric tests used as needed for comparisons, with paired tests used within subjects for repeated measures. The relationship between fluid biomarkers and neuroimaging markers will be assessed, according to previously used approaches.[27] We will use linear regression to look at the relationship between blood biomarker levels and continuous outcomes, or logistic regression for binarised outcomes

such favourable/unfavourable recovery on the Glasgow Outcome Scale-Extended (GOSE) at 6 or 12 months. Cerebral microdialysis marker levels will be described over time and cross-compared to blood biomarker results, neuroimaging markers of axonal injury such as fractional anisotropy, and clinical outcomes, per Magnoni *et al*[16]

### Neuroimaging

MRI and CT data will be centralised at ICL. MRI reporting will be performed by neuroradiologists in London to allow comparison across sites, while CTs are reported locally. Individualised 'masks' will be manually drawn on structural MRI images to allow focal lesions to be excluded from later analyses, for each scanning session, for each study participant.

The following analyses will be performed:

1. Diffusion tensor imaging—a tract-based spatial statistics approach will be used to generate voxelwise maps of fractional anisotropy (FA) using well-established approaches in the TBI setting.[5] These will be compared between groups, allowing a description of axonal injury multiple levels including whole white matter, voxelwise and tractwise, as per Jolly *et al*.[28] To account for cross-site variability related to scanner, this will be included as a nuisance covariate in analyses. For cross-sectional diffusion imaging, for each scanning site, patient FA data will be normalised via a z-scoring approach to local controls carefully matched for age and sex. Z-scored FA data can be more reliably combined across sites, and larger groupwise analyses performed. Additionally, we will explore the use of a voxelwise algorithm designed to harmonise diffusion data across sites.[29]

2. Longitudinal atrophy assessment—we will use approaches previously established in the setting of moderate-to-severe TBI to generate individualised maps of brain volume change over time (Jacobian Determinant 'JD' atrophy rate maps) from serial T1 images.[26] Diffuse brain volume changes, such as related to neurodegeneration due to injury, will be separated from the resolution of focal pathologies by the use of focal lesion masks, such that JD values in non-lesioned areas can be sampled and assessed. As each individual acts as their own 'control' longitudinally, we do anticipate a significant influence of scanner on atrophy rate assessment, and will quantify this effect using hierarchical partitioning of variance on linear regression.

3. Resting state fMRI—we will assess the effect of injury on brain network function cross sectionally and longitudinally using approaches previously applied to TBI, relating structural and functional connectivity.[30]

4. CT—we will apply several different approaches to test whether CT neuroimaging data can assist in outcome. We will test whether an automatic lesion segmentation algorithm trained in patients with TBI produces outputs which reflect known clinical features and outcomes such as the GOSE.[31] We will train a multimodal neural network to take in scans at multiple timepoints

within an individual's hospital admission, with auxiliary patient data, to predict outcome. We will assess supervised learning and will investigate methods to increase the model interpretability, such as heatmaps.

## Data management

Data and all appropriate documentation will be stored for a minimum of 10 years after the completion of the study. This is to enable subsequent analyses with new network analysis techniques as they are developed. All study data will be stored, analysed and published in anonymised format. Fluid biomarkers will be stored at UCL and neuroimaging data at ICL.

As part of a large European multicentre study of clinical outcomes following TBI, pseudonymised clinical data will be shared with the CREACTIVE network. The research programme of CREACTIVE has a strong emphasis on data collection, aimed at obtaining all available relevant clinical data of the patients admitted to the major trauma centres in the CREACTIVE network, in this instance, all patients admitted to participating MTCs in London. Data to be collected on these patients will include pseudonymised follow-up data, related to postacute patient care and patient outcomes. The pseudonymised data will be uploaded to an encrypted database on a network which is accessible to the CREACTIVE team. This database has full https encryption with a certificate from an official authority. Both the database and the application into which the data are loaded are run from dedicated servers. The data collected for this component of the study will be centrally stored by the CREACTIVE team for 20 years from the closing date of the study. At that time point, all data collected for this study will be definitively deleted. If these data are considered at that time to be particularly relevant for scientific purposes, a specific authorisation to continue their use will be issued to the competent local/national relevant ethical boards/bodies.

## ETHICS AND DISSEMINATION
### Ethical considerations

The relevant ethical approvals have been granted by the following ethics committees: in London, by the Camberwell St Giles Research Ethics Committee; in Policlinico (Milan), by the Comitato Etico Milano Area 2; in Niguarda (Milan), by the Comitato Etico Milano Area 3; in Careggi (Florence), by the Comitato Etico Regionale per la Sperimentazione Clinica della Regione Toscana, Sezione area vasta centro; in Trento, by the Trento Comitato Etico per le Sperimentazioni Cliniche, Azienda Provinciale per i Servizi Sanitari della Provincia autonoma di Trento; in Lausanne, by the Commission cantonale d'éthique de la recherche sur l'être humain; in Ljubljana, by the National Medical Ethics Committee at the Ministry of Health of the Republic of Slovenia.

Periods of PTA or low consciousness level are commonly observed after TBI, and during this period, patients may not have mental capacity give valid informed consent. An important element of the proposed work cannot otherwise be answered by only studying patients with capacity.

The proposed investigation methods are routinely used in the assessment of patients with TBI in an intensive care setting and are safe, as long as normal safety precautions are taken in the case of MRI. We have used this safely to investigate the causes of poor clinical outcome after TBI.

At the time of initial contact by the clinical team an assessment of the patient's capacity will be made in relation to entering the study in accordance with national legislation. If the patient is not found to have capacity then they will be enrolled, if appropriate, following the national rules. Patients will not be coerced if assent is not given. If during the study the patient subsequently objects to participation in a portion of the study she/he will be withdrawn. Advanced statements regarding participation in research will be sought and respected.

When the patient is no longer in PTA or a state of impaired consciousness, we will repeat the consent procedure and the patient will be free to withdraw from the study or continue participation.

Our study will involve negligible risk to the patient and will not significantly interfere with freedom of action or privacy or be unduly invasive or restrictive. The primary inconvenience to the patient is to cooperate with an MRI scan which can take up to 60 min and, for some patients, be noisy, uncomfortable and claustrophobic. We will consider using light oral sedation such as benzodiazepines (diazepam) if this is judged to be helpful. There is also a risk to patients if they have ferromagnetic metal on their person and all the usual MRI safety checks and procedures will be undertaken to minimise this risk. Uncooperative and unconscious patients at the time of MRI (eg, intensive care treated patients) will receive sedation and will be monitored for vital signs under the surveillance of the treating physician and according to local protocols for MRI in critically ill patients.

## Dissemination

The study findings will be disseminated to patients, healthcare professionals, academics and policy-makers including through presentation at conferences and peer-reviewed publications. Data will be shared with approved researchers to provide further insights for patient benefit.

## Patient and public involvement

Patients and research participants were involved in the formulation of our programme via regular participant involvement events at the sponsoring institution (ICL).

**Author affiliations**
[1]Department of Brain Sciences, Imperial College London, London, UK
[2]UK DRI Centre for Care Research and Technology, London, UK
[3]Public Health, IRCCS-'Mario Negri' Institute for Pharmacological Research, Ranica, Italy
[4]Department of Anesthesia and Intensive Care, Santa Chiara Hospital, Trento, Italy
[5]Department of Intensive Care Medicine, CHUV Lausanne University Hospital, Lausanne, Switzerland
[6]Department of Neurodegenerative Disease, UCL Queen Square Institute of Neurology, London, UK

[7]Department of Psychiatry and Neurochemistry, Institute of Neuroscience and Physiology, University of Gothenburg Sahlgrenska Academy, Mölndal, Sweden
[8]UCL Queen Square Institute of Neurology, London, UK
[9]Neurorianimazione, ASST Grande Ospedale Metropolitano Niguarda, Milano, Italy
[10]Department of Experimental and Clinical Sciences, Careggi University Hospital, University of Firenze, Florence, Italy
[11]Clinical Department of Anaesthesiology and Intensive Therapy, University Medical Center, Ljubljana, Slovenia

**Acknowledgements** We thank the patients who contributed to design of our research programme via regular participation events at Imperial College London. We thank the patients who contributed to design of our research programme via regular participation events at Imperial College London.

**Contributors** NSNG, KAZ and DJS drafted the manuscript. GB, SM, MO, HZ, FM, DN, AH, JMF, AC, EF, EG, SA-M, PG and AB critically reviewed and revised it. All authors contributed to the design, intellectual content and approved the final version.

**Funding** The study is funded by the ERA-NET Neuron Cofund (grant MR/R004528/1), part of the European Research Projects on External Insults to the Nervous System call, within the Horizon 2020 funding framework. The work was supported by an Alzheimer's Research UK Clinical Research Fellowship awarded to NSNG, and by the UK Dementia Research Institute (DRI) Care Research and Technology Centre (NG/DJS), an National Institute of Health Research (NIHR) Professorship (NIHR-RP-011-048) awarded to DJS. and by the NIHR Clinical Research Facility and Biomedical Research Centre (BRC) at Imperial College Healthcare NHS Trust. The research was also supported the Medical Research Council through a Clinician Scientist Fellowship awarded to DJS.

**Competing interests** HZ has served at advisory boards for Eli Lilly, Roche Diagnostics and Pharmasum Therapeutics, and is a cofounder of Brain Biomarker Solutions in Gothenburg AB, a GU Ventures-based platform company at the University of Gothenburg. The other authors have no potential conflicts to declare.

**Patient consent for publication** Not required.

**Provenance and peer review** Not commissioned; externally peer reviewed.

**ORCID iDs**
Neil Samuel Nyholm Graham http://orcid.org/0000-0002-0183-3368
Arturo Chieregato http://orcid.org/0000-0001-6690-8597
Elena Garbero http://orcid.org/0000-0003-4902-0144

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
