## [Reviewer comments · BMJ Open]

ARTICLE DETAILS

TITLE (PROVISIONAL)	A multi-centre longitudinal study of fluid and neuroimaging BIOmarkers of AXonal injury after Traumatic Brain Injury: the BIO-AX-TBI study protocol
AUTHORS	Graham, Neil; Zimmerman, Karl; Bertolini, Guido; Magnoni, Sandra; Oddo, Mauro; Zetterberg, Henrik; Moro, Federico; Novelli, Deborah; Heslegrave, Amanda; Chierigato, Arturo; Fainardi, Enrico; Fleming, Joanne; Garbero, Elena; Abed-Maillard, Samia; Gradisek, Primoz; Bernini, Adriano; Sharp, David

VERSION 1 – REVIEW

REVIEWER	Alfonso Lagares Hospital Universitario 12 de Octubre, Universidad Complutense de Madrid, Instituto de Investigación Imas12
REVIEW RETURNED	27-Jul-2020

GENERAL COMMENTS	This is a protocol of a multi center observational study to assess the value of blood biomarkers in defining diffuse atonal injury or traumatic atonal injury(TAI) and its role in TBI prognosis. The paper describes the protocol and the different work packages in which different centers will contribute to the common goal of ascertain the value of biomarkers in detecting TAI. Authors should address several questions regarding the protocol in order to clarify it: -Several work packages are described including a more general 4 WP. There is no center fulfilling this last W: will it be a separate study? As there is no clear guideline for this WP will it have a separate ethics approval? How many blood samples will be drawn for this WP?-From the basis of sample size it seems recruitment for the study has been already completed or even more than needed patients have been included already in the study making late its publication as a protocol.-Sample size is calculated in the basis of DTI as a prognostic factor. It is not clear then which is the main objective of the research: The value of DTI to define prognosis or the relationship of brain biomarkers and the presence of DTI defined damage and assess prognosis? As per the title of the protocol it would be expected that sample size would be defined by biomarker assessment not DTI. Some explanation regarding the second paragraph is also needed...which are the groups that are cited in the estimation of sample size for the detection of atrophy? Although atrophy is mentioned in the study there is no information regarding which method will be used to measure atrophy.-Statistical analysis: there is no description of what measure will be used for the analyst is of the primary and secondary outcomes of the study. Authors should state which analysis will be performed
--

	at least for the primary outcome of the study. If work is done in different WP at least they should describe how this different WP will interact. -Study limitations: There is a lack of discussion regarding limitations. Regarding DTI analysis the use of different MR hardware and software could be a major limitation as FA values are severely affected by this. Will authors have some form of corrective strategy? They should include this as a limitation. -There is no information regarding how DTI, fMR or atrophy will be assessed and used along the study....will there be a central reading of these characteristics? And of MR or CT findings?
--	--

REVIEWER	Regina C. Armstrong Uniformed Services University of the Health Sciences, USA
REVIEW RETURNED	01-Aug-2020

GENERAL COMMENTS	The natural history study protocol reported by Dr. Graham and colleagues examines blood, microdialysis, and neuroimaging biomarkers as surrogate outcome measures for patients with moderate-severe traumatic brain injury (TBI). This longitudinal study is conducted from acute (10 days) through 1 year post-injury time points across multiple sites. The protocol is very timely in that the biomarker methods have matured sufficiently to be selected appropriately based on strong prior data in the literature to support the analysis plans and time points. Successful completion of the protocol should result in data that can be used to significantly improve the design of clinical trials to assess therapeutics in patients with TBI that target axonal damage, which is a major pathological component underlying persistent symptoms and long term disability. Minor clarifications should be considered in this report of the study protocol.  1. This report is for an ongoing longitudinal study that is stated as having reached the recruitment goal of at least 250 participants in that 311 participants have been recruited. Please clarify on Page 10 whether the 311 participants is as of February 2020 (line 2) or June 2020 (figure legend). 2. Please clarify the post-injury interval for recruitment, and intervals between blood samples. The patients are recruited at the acute post-injury stage. At least 24 hours is given to consider the enrollment information (Page 11 line 12). Blood samples are collected twice within the first 10 days after injury (Page 11 line 60). 3. For the purpose of reporting the study plans as a manuscript, the report would be more useful if the methods to be used were more clearly explained for the analyses of blood for the named candidate proteins. This would allow the readers to connect to the relevant literature on those analysis techniques to appreciate the sensitivity and specificity expected. Similarly, for the overall approach of the genetics, the reader would gain from knowing the type of genetic analysis planned.
---

REVIEWER	Emily Grossner The Pennsylvania State University State College, Pennsylvania, USA
REVIEW RETURNED	10-Aug-2020

GENERAL COMMENTS	This is a very detailed study protocol examining the role of potential biomarkers of axonal injury in predicting outcome following TBI. The authors have clearly stated the aims of the study, along with detailed work packages and locations that will address each of the aims. This is important work that will further the understanding of the use of biomarkers in TBI. Statistical methods and analyses were not included in this study protocol. A proposed analysis section might include statistical prediction models used to predict outcome from biomarkers. Additionally, it is indicated that machine learning will be used with work package 4. A description of the machine learning algorithm would allow for statistical replication. Lastly, a more thorough description of the proposed sample size of each location would help to ensure a relatively equal distribution at patients at each site. Due to the differences in MRI scanners at each of these sites, a relatively equal number of patients at each site would help to assure that any imaging findings are not driven by these scanner differences. To this point as well, a description of measures that will be taken to ensure data fidelity from differing scanners would be helpful (i.e., signal-to-noise ratio data, controls for different scanner type and strength). Statistical methods and analyses were not included in this study protocol. A proposed analysis section might include statistical prediction models used to predict outcome from biomarkers. Additionally, it is indicated that machine learning will be used with work package 4. A description of the machine learning algorithm would allow for statistical replication. Lastly, a more thorough description of the proposed sample size of each location would help to ensure a relatively equal distribution at patients at each site. Due to the differences in MRI scanners at each of these sites, differences in findings would not want to be attributed to differing technologies. To this point as well, a description of measures that will be taken to ensure data fidelity from differing scanners would be helpful (i.e., signal-to-noise ratio data, controlling for scanner type, etc.).
---

VERSION 1 – AUTHOR RESPONSE

Reviewer: 1. Reviewer Name Alfonso Lagares. Institution and Country Hospital Universitario 12 de Octubre, Universidad Complutense de Madrid, Instituto de Investigación Imas12 Please state any competing interests or state 'None declared': None declared

Please leave your comments for the authors below. This is a protocol of a multi center observational study to assess the value of blood biomarkers in defining diffuse atonal injury or traumatic atonal injury(TAI) and its role in TBI prognosis. The paper describes the protocol and the different work packages in which different centers will contribute to the common goal of ascertain the value of biomarkers in detecting TAI. Authors should address several questions regarding the protocol in order to clarify it: Several work packages are described including a more general 4 WP. There is no center fulfilling this last W: will it be a separate study? As there is no clear guideline for this WP will it have a separate ethics approval? How many blood samples will be drawn for this WP?

We thank the reviewer for their comments on the protocol. We agree that the description of WP4 could be clearer and have revised page 9, line 17 to reflect this. No separate ethical approvals are required for this analysis-only work package, which forms part of the main BIOAXTBI study.

Work package 4 is an analysis-only package involving comparing our data (WP1-3) to a broader sample of TBI patients collected within the related CREATIVE project, including neuroimaging data. This work package does not involve collection of new data or biological samples. Two main analyses are planned: first, to compare clinical features, biomarkers and outcomes in patients in the BIOAXTBI cohort with the broader CREATIVE population, where these are in-common, and establish how well findings from WP1-3 generalise in the larger group. Second, a machine learning analysis will be performed to establish whether CT brain imaging data in this large cohort can reliably predict outcomes.

From the basis of sample size it seems recruitment for the study has been already completed or even more than needed patients have been included already in the study making late its publication as a protocol.

We apologise that this was unclear and have amended the manuscript as follows (page 7 line 13):

Our targets are to recruit at least 250 patients in total, including 40 with invasive cerebral microdialysis. As of June 2020, 311 patients after moderate-severe TBI have been recruited into the study (Figure 1), with recruitment ongoing in order to meet our target in the cerebral microdialysis group.

Sample size is calculated in the basis of DTI as a prognostic factor. It is not clear then which is the main objective of the research: The value of DTI to define prognosis or the relationship of brain biomarkers and the presence of DTI defined damage and assess prognosis? As per the title of the protocol it would be expected that sample size would be defined by biomarker assessment not DTI. Some explanation regarding the second paragraph is also needed...which are the groups that are cited in the estimation of sample size for the detection of atrophy? Although atrophy is mentioned in the study there is no information regarding which method will be used to measure atrophy.

We agree that more detail would help in this regard and have revised page 14, line 3 as suggested:

The primary outcome measures are (1) change in diffusion tensor MRI measures over time measured using fractional anisotropy; (2) Brain atrophy rates using JD atrophy rates; (3) Change in levels of fluid biomarkers in blood (4) Change in levels of fluid biomarkers in cerebral fluid.

We estimate that a minimum of 140 patients will be necessary to test the contribution of DTI to prognostic modelling (type-1 error=0.05, power=0.95). Loss to follow-up is an important but unpredictable element and we have allowed for up to a further 25-30% loss to follow-up by aiming to enrol 250 patients in the study. Our estimates indicated that a minimum of 12 imaging controls will be necessary for each centre.

To assess atrophy progression after TBI in work package two, our calculations indicate a sample size of 10 per group is needed to detect a group difference in brain atrophy over 6 months with effect size of $f = 0.32$, assuming 95% and a correlation between repeated measures = 0.9. Longitudinal brain atrophy rates will be calculated using the JD measure of atrophy rate generated using SPM 12 (UCL).²⁶

Preliminary data suggests that moderate-severe TBI induces large changes in blood biomarker concentrations (eg. S100B, NFL). We estimate that the numbers required to show between group differences and relationships to outcome will be smaller than for DTI (1) or brain atrophy rates (2), on whose measures the study is primarily powered.

We have based the WP3 sample size (n=40) on previous similar microdialysis studies that have used 15-20 subjects and shown correlations between biomarker and imaging measures. With a sample size of 40 we should be adequately powered for this analysis, as well as being able to take into account the additional variability related to different centres.

Statistical analysis: there is no description of what measure will be used for the analysis of the primary and secondary outcomes of the study. Authors should state which analysis will be performed at least for the primary outcome of the study. If work is done in different WP at least they should describe how this different WP will interact.

We have now added a 'statistical analysis plan' section to the manuscript as suggested (page 14, line 30)

Statistical analysis plan

Fluid biomarkers

Blood biomarker trends will be described and compared between groups, as well as within individuals longitudinally. Distributions of variables will be assessed for normality and parametric/non-parametric tests used as needed for comparisons, with paired tests used within subjects for repeated measures. The relationship between fluid biomarkers and neuroimaging markers will be assessed, according to previously used approaches.²⁷ We will use linear regression to look at the relationship between blood biomarker levels and continuous outcomes, or logistic regression for binarised outcomes such as favourable / unfavourable recovery on the GOSE at 6 or 12 months. Cerebral microdialysis marker levels will be described over time and cross-compared to blood biomarker results, neuroimaging markers of axonal injury such as fractional anisotropy, and clinical outcomes, per Magnoni et al.¹⁶

Neuroimaging

MRI and CT data will be centralised at Imperial College London. MRI reporting will be performed by neuroradiologists in London to allow comparison across sites, while CTs are reported locally. Individualised 'masks' will be manually drawn on structural MRI images to allow focal lesions to be excluded from later analyses, for each scanning session, for each study participant.

The following analyses will be performed:

i) Diffusion tensor imaging – a tract based spatial statistics approach will be used to generate voxelwise maps of fractional anisotropy using well-established approaches in the TBI setting.⁵ These will be compared between groups, allowing a description of axonal injury multiple levels including whole white matter, voxelwise and tractwise, as per Jolly et al.²⁸ To account for cross-site variability related to scanner, this will be included as a nuisance covariate in analyses. For cross-sectional diffusion imaging, for each scanning site, patient FA data will be normalised via a z-scoring approach to local controls carefully matched for age and sex. Z-scored FA data can be more reliably combined across sites, and larger group-wise analyses performed. Additionally, we will explore the use of a voxelwise algorithm designed to harmonise diffusion data across sites.²⁹

ii) Longitudinal atrophy assessment – we will use approaches previously established in the setting of moderate-severe TBI to generate individualised maps of brain volume change over time

(‘JD atrophy rate’ maps) from serial T1 images.²⁶ Diffuse brain volume changes, such as related to neurodegeneration due to injury will be separated from the resolution of focal pathologies by the use of focal lesion masks, such that JD values in non-lesioned areas can be sampled and assessed. As each individual acts as their own ‘control’ longitudinally, we do anticipate a significant influence of scanner on atrophy rate assessment, and will quantify this effect using hierarchical partitioning of variance on linear regression.

iii) Resting state functional MRI – we will assess the effect of injury on brain network function cross sectionally and longitudinally using approaches previously applied to TBI, relating structural and functional connectivity.³⁰

iv) CT – we will apply several different approaches to test whether CT neuroimaging data can assist in outcome. We will test whether an automatic lesion segmentation algorithm trained in TBI patients produces outputs which reflect known clinical features and outcomes such as the GOSE.³¹ We will train a multimodal neural network to take in scans at multiple timepoints within an individual’s hospital admission, with auxiliary patient data, to predict outcome. We will assess supervised learning and will investigate methods to increase the model interpretability, such as heatmaps.

Study limitations: There is a lack of discussion regarding limitations. Regarding DTI analysis the use of different MR hardware and software could be a major limitation as FA values are severely affected by this. Will authors have some form of corrective strategy? They should include this as a limitation. -There is no information regarding how DTI, fMR or atrophy will be assessed and used along the study....will there be a central reading of these characteristics? And of MR or CT findings?

We agree that more detail would help here and have clarified this in the above statistical analysis plan section under ‘Neuroimaging Analysis’ - page 15, line 13. On page 15 line 25 we set out how we will manage between-scanner variability particularly in respect of DTI measures.

Reviewer: 2

Reviewer Name. Regina C. Armstrong

Institution and Country. Uniformed Services University of the Health Sciences, USA; Please state any competing interests or state ‘None declared’ None declared

Please leave your comments for the authors below

The natural history study protocol reported by Dr. Graham and colleagues examines blood, microdialysis, and neuroimaging biomarkers as surrogate outcome measures for patients with moderate-severe traumatic brain injury (TBI). This longitudinal study is conducted from acute (10 days) through 1 year post-injury time points across multiple sites. The protocol is very timely in that the biomarker methods have matured sufficiently to be selected appropriately based on strong prior data in the literature to support the analysis plans and time points. Successful completion of the protocol should result in data that can be used to significantly improve the design of clinical trials to assess therapeutics in patients with TBI that target axonal damage, which is a major pathological

component underlying persistent symptoms and long term disability. Minor clarifications should be considered in this report of the study protocol.

1. This report is for an ongoing longitudinal study that is stated as having reached the recruitment goal of at least 250 participants in that 311 participants have been recruited. Please clarify on Page 10 whether the 311 participants is as of February 2020 (line 2) or June 2020 (figure legend).

We thank the reviewer for their comments. We have amended the manuscript as follows (page 7 line 13) to address this:

Our targets are to recruit at least 250 patients in total, including 40 with invasive cerebral microdialysis. As of June 2020, 311 patients after moderate-severe TBI have been recruited into the study (Figure 1), with recruitment ongoing in order to meet our target in the cerebral microdialysis group.

2. Please clarify the post-injury interval for recruitment, and intervals between blood samples. The patients are recruited at the acute post-injury stage. At least 24 hours is given to consider the enrollment information (Page 11 line 12). Blood samples are collected twice within the first 10 days after injury (Page 11 line 60).

We have amended page 8, line 7, to clarify the timing of recruitment into the study:

Individuals will have the opportunity to ask questions about the study. Questions will be asked to assess the individual's suitability for the study. Potential participants will be asked to consider the request for a period of 24 hours prior to recruitment into the study. If the potential participant is suitable for the study, informed consent will be obtained. Our goal is to recruit patients as early after trauma as is practical, and ideally within ten days of injury in order to facilitate acute blood biomarker assessment.

3. For the purpose of reporting the study plans as a manuscript, the report would be more useful if the methods to be used were more clearly explained for the analyses of blood for the named candidate proteins. This would allow the readers to connect to the relevant literature on those analysis techniques to appreciate the sensitivity and specificity expected. Similarly, for the overall approach of the genetics, the reader would gain from knowing the type of genetic analysis planned.

We have added the following sentence to page 12, line 3, to clarify this.

Fluid biomarker analyses will be performed at UCL using a digital ELISA technique, using a Quanterix Simoa analyser to provide ultrasensitive assessment of concentration.

The following sentence has been added to page 12 line 12 to describe the genetic analysis plan:

Genetic analyses may include APOE genotype assessment and/or generation of an individualised polygenic risk score for neurodegeneration, using a microarray targeting SNPs related to neurodegenerative disease.²³

Reviewer: 3

Reviewer Name Emily Grossner. ; Institution and Country: The Pennsylvania State University State College, Pennsylvania, USA; Please state any competing interests or state 'None declared': None declared

Please leave your comments for the authors below

This is a very detailed study protocol examining the role of potential biomarkers of axonal injury in predicting outcome following TBI. The authors have clearly stated the aims of the study, along with detailed work packages and locations that will address each of the aims. This is important work that will further the understanding of the use of biomarkers in TBI.

Statistical methods and analyses were not included in this study protocol. A proposed analysis section might include statistical prediction models used to predict outcome from biomarkers. Additionally, it is indicated that machine learning will be used with work package 4. A description of the machine learning algorithm would allow for statistical replication.

We are grateful to the reviewer for their comments on the paper and have now added a detailed statistical analysis plan, including information describing the machine learning CT analysis (page 14, line 30):

Statistical analysis plan

Fluid biomarkers

Blood biomarker trends will be described and compared between groups, as well as within individuals longitudinally. Distributions of variables will be assessed for normality and parametric/non-parametric tests used as needed for comparisons, with paired tests used within subjects for repeated measures. The relationship between fluid biomarkers and neuroimaging markers will be assessed, according to previously used approaches.²⁷ We will use linear regression to look at the relationship between blood biomarker levels and continuous outcomes, or logistic regression for binarised outcomes such as favourable / unfavourable recovery on the GOSE at 6 or 12 months. Cerebral microdialysis marker levels will be described over time and cross-compared to blood biomarker results, neuroimaging markers of axonal injury such as fractional anisotropy, and clinical outcomes, per Magnoni et al.¹⁶

Neuroimaging

MRI and CT data will be centralised at Imperial College London. MRI reporting will be performed by neuroradiologists in London to allow comparison across sites, while CTs are reported locally. Individualised 'masks' will be manually drawn on structural MRI images to allow focal lesions to be excluded from later analyses, for each scanning session, for each study participant.

The following analyses will be performed:

i) Diffusion tensor imaging – a tract based spatial statistics approach will be used to generate voxelwise maps of fractional anisotropy using well-established approaches in the TBI setting.⁵ These will be compared between groups, allowing a description of axonal injury multiple levels including whole white matter, voxelwise and tractwise, as per Jolly et al.²⁸ To account for cross-site variability related to scanner, this will be included as a nuisance covariate in analyses. For cross-sectional diffusion imaging, for each scanning site, patient FA data will be normalised via a z-scoring approach to local controls carefully matched for age and sex. Z-scored FA data can be more reliably combined across sites, and larger group-wise analyses performed. Additionally, we will explore the use of a voxelwise algorithm designed to harmonise diffusion data across sites.²⁹

ii) Longitudinal atrophy assessment – we will use approaches previously established in the setting of moderate-severe TBI to generate individualised maps of brain volume change over time

(‘JD atrophy rate’ maps) from serial T1 images.²⁶ Diffuse brain volume changes, such as related to neurodegeneration due to injury will be separated from the resolution of focal pathologies by the use of focal lesion masks, such that JD values in non-lesioned areas can be sampled and assessed. As each individual acts as their own ‘control’ longitudinally, we do anticipate a significant influence of scanner on atrophy rate assessment, and will quantify this effect using hierarchical partitioning of variance on linear regression.

iii) Resting state functional MRI – we will assess the effect of injury on brain network function cross sectionally and longitudinally using approaches previously applied to TBI, relating structural and functional connectivity.³⁰

iv) CT – we will apply several different approaches to test whether CT neuroimaging data can assist in outcome. We will test whether an automatic lesion segmentation algorithm trained in TBI patients produces outputs which reflect known clinical features and outcomes such as the GOSE.³¹ We will train a multimodal neural network to take in scans at multiple timepoints within an individual’s hospital admission, with auxiliary patient data, to predict outcome. We will assess supervised learning and will investigate methods to increase the model interpretability, such as heatmaps.

Lastly, a more thorough description of the proposed sample size of each location would help to ensure a relatively equal distribution at patients at each site. Due to the differences in MRI scanners at each of these sites, a relatively equal number of patients at each site would help to assure that any imaging findings are not driven by these scanner differences. To this point as well, a description of measures that will be taken to ensure data fidelity from differing scanners would be helpful (i.e., signal-to-noise ratio data, controls for different scanner type and strength).

We agree that this is a key consideration in our multi-site study and have now provide more detailed information on the planned approach in the statistical analysis plan (see above). With age-matched controls at each site, using a z-scoring approach where each patient’s FA map is normalised to their own (local) control population, we expect to be able to mitigate the effect of scanner variability across the group.

VERSION 2 – REVIEW

REVIEWER	Alfonso Lagares Hospital Universitario 12 de Octubre, Universidad Complutense de Madrid, Instituto de Investigación imas12
REVIEW RETURNED	09-Sep-2020

GENERAL COMMENTS	Thanks for your response to the comments made on your paper. I believe now the protocol has gained in clarity and the reader will be able to better understand the analysis that are planned.
---

REVIEWER	Regina C. Armstrong, PhD Uniformed Services University of the Health Sciences
REVIEW RETURNED	13-Sep-2020

GENERAL COMMENTS	The authors have addressed the concerns of the prior reviews.
---

REVIEWER	Emily Grossner The Pennsylvania State University, State College, Pennsylvania, USA
REVIEW RETURNED	15-Sep-2020

GENERAL COMMENTS	This protocol examines potential biomarkers of axonal injury in predicting outcome following TBI. Prior to the revisions, the authors clearly stated that aims of the study and detailed the work packages that would address each of the aims. The authors have now provided a detailed section on planned statistical analyses that address all of the analyses to be conducted using blood-based biomarkers and neuroimaging measures. Within this section, the authors have clarified how they will address data harmonization in imaging analyses by using standardized z-score values and within-subject longitudinal analyses. This section clarifies the questions the reviewers had pertaining to data analysis.
--